# Cortical and medullary oxygenation evaluation of kidneys with renal artery stenosis by BOLD-MRI

**Long Zhao[1], Guoqi Li[2], Fanyu Meng[3], Zhonghua Sun[4], Jiayi Liu[1] ***

**1** Department of Radiology, Beijing Anzhen Hospital, Beijing, China, **2** Energy conservation and environmental protection division, Aerospace HIWING Security Technology Engineering Co., Ltd, Beijing, China, **3** International Cooperation Office, Beijing Anzhen Hospital, Beijing, China, **4** Discipline of Medical Radiation Science, Curtin Medical School, Curtin University, Perth, Australia

* ljy76519@163.com

**Data Availability Statement:** We are glad to share all relevant data, but some of data that support the findings of this study are not publicly available due to privacy or ethical restrictions. Because raw data contain patients' names, IDs, genders, ages, medical histories, diseases etc. Meanwhile, the raw

## Abstract

### Aim

Blood oxygen level–dependent magnetic resonance imaging (BOLD-MRI) can measure deoxyhemoglobin content. This study aims to evaluate the capacity of BOLD-MRI, which is possible to evaluate the oxygenation state of kidneys with renal artery stenosis (RAS).

### Materials and methods

We performed BOLD-MRI for 40 patients with RAS and for 30 healthy volunteers. We then performed post-scan processing and analysis of manually drawn regions of interest to determine R2* values (relaxation rates) for the renal cortex and medulla. We compared R2* values in patients with RAS with those in the control group, and also compared these values for subgroups with varying degrees of stenosis.

### Results

Medulla R2* values were higher than cortex R2* values in the control group. There was no significant difference in R2* values for different segments (upper, middle, lower) of the kidneys. Both cortex and medulla R2* values in patients with RAS were significantly higher than corresponding R2* values in the control group ($P < 0.05$), and BOLD-MRI was more sensitive to changes in the R2* values in the medulla than in the cortex. Among different subgroups in the RAS group, the medulla R2* values were significantly higher in kidneys with severe stenosis than in those with no obvious obstruction, mild stenosis, or moderate stenosis ($P < 0.05$).

### Conclusion

BOLD-MRI is an effective, noninvasive method for evaluating kidney oxygenation, which is important for proper treatment in RAS. It is sufficiently sensitive for detecting medulla ischemia and anoxia of the kidneys.

data of this study cannot be shared at this time as the data also forms part of an ongoing study. Data requests could connect to the academic research office of Bejing Anzhen hospital; E-mail: anzhenkjc@163.com.

**Funding:** The authors received no specific funding for this work.

**Competing interests:** The authors have declared that no competing interests exist.

## Introduction

Magnetic resonance imaging (MRI) offers outstanding soft-tissue resolution and does not use ionizing radiation, and this has made it the optimal choice of imaging method for revealing anatomical structures. With increasing research on the uses of MRI in physics, chemistry, biology, and medicine, more and more researchers have turned to functional MRI to obtain information about both anatomical structure and anatomical function. Functional MRI was first applied in research on neurophysiology, mainly in the study of the visual and functional cortex [1]. The most widely used technique in such studies is blood oxygen level–dependent MRI (BOLD-MRI) [1]. In recent years, the application of BOLD-MRI in kidney assessment has received increasing attention from clinical researchers. Studies of renal BOLD-MRI focused on renal ischemic disease, diabetic nephropathy, chronic kidney disease, renal failure, renal transplantation, and renal space-occupying lesions [2, 3]. BOLD-MRI provides both conventional anatomical information and functional information that can have clinical significance in the early detection of functional changes in diseased local tissues, allowing early diagnosis and treatment.

Renal artery stenosis (RAS) is common in artery disease. Although many patients have no obvious clinical symptoms, diagnosis and treatment of RAS has still been the subject of clinical attention because RAS can decrease renal blood perfusion, leading to activation of the renin angiotensin aldosterone system. RAS could also lead to a series of life-threatening problems, including secondary hypertension, ischemic nephropathy, renal atrophy, renal failure, and congestive heart failure [4]. Many imaging methods are currently used to examine patients with RAS, including color Doppler ultrasound, computed tomography angiography(CTA), and magnetic resonance angiography (MRA). These imaging modalities can show the degree of RAS directly or indirectly, however, direct examination of the functioning of the renal parenchyma and microcirculation by a sensitive technique is rarely performed, although it is necessary to determine proper treatment in RAS. Thus, we conducted this study to investigate the effectiveness of BOLD-MRI for detecting RAS in these renal structures. We performed post-scan processing to determine $R2^*$ values (relaxation rates) in the cortex and medulla (1/second) in order to assess the kidney oxygenation state. We then compared $R2^*$ values for study participants with RAS against the $R2^*$ values for a control group. Further, we compared and analyzed $R2^*$ values for kidneys with different degrees of RAS to investigate the changes and characteristics of renal ischemia and hypoxia in patients with RAS.

## Materials and methods

### Patients

Between October 2017 and December 2018, we enrolled 40 patients with RAS into a patient study group. Clinical data and general information were recorded for all patients. In addition, each patient underwent renal artery examination by ultrasound. We assessed the degree of RAS using a 5-point ordinal scale: 0 = no obvious stenosis, 1 = mild stenosis (<50% stenosis), 2 = moderate stenosis (50–75% stenosis), 3 = severe stenosis (>75% stenosis), and 4 = total occlusion [5]. When the degree of stenosis was difficult to judge, a joint decision was made by 2 experienced radiologists (over 10 years of experience). Because renal arterial occlusion causes kidney atrophy [6], the resulting unclear demarcation between the renal cortex and the medulla limits the usefulness of BOLD-MRI for evaluation [7]. Therefore, patients with the degree of RAS being 4 (renal artery occlusion) were excluded from the study. Further, since diabetic nephropathy can affect renal $R2^*$ values, we also excluded patients with RAS who also had diabetes mellitus [8].

**Table 1. Subgroups of patients with different degrees of renal artery stenosis.**

| Degree of stenosis | No. of kidneys | Average age of patients (years) | Gender (male/female) |
|---|---|---|---|
| 0 | 17 | 52.13 ± 16.84 | 12/5 |
| 1 | 11 | 62.50 ± 7.06 | 5/6 |
| 2 | 14 | 60.67 ± 17.66 | 7/7 |
| 3 | 25 | 59.20 ± 14.75 | 15/10 |

Other exclusion criteria included the presence of severe cardiac and/or pulmonary insufficiency, contraindications to MRI, recent infection, and recent use of paramagnetic drugs or diuretics. This resulted in a total of 36 patients (21 men and 15 women with an average age of 57.63 ± 15.42 years, age range: 23–78 years) (72 kidneys) in the study group, 30 of whom had hypertension and 6 of whom had normal blood pressure. Of 36 patients, 5 had unilateral renal atrophy, thus we eliminated the data leaving $R2^*$ values for 67 kidneys to be analyzed. We divided those 67 kidneys into 4 groups (Table 1): RAS with no obvious stenosis (17 kidneys), mild stenosis (11), moderate stenosis (14), and severe stenosis (25).

At the same time, we included a control group consisting of 30 volunteers (21 men and 9 women) without hypertension, diabetes, or urinary system diseases without recently taking any medications (Table 2). The average age of control-group participants was 57.38 ± 14.50 years (range, 30–78 years). We collected a total of 60 sets of renal BOLD-MRI data from the control group.

This study was approved by the human research ethics committee of our hospital. All patients with RAS and all healthy volunteers were provided with detailed information regarding test purpose and study methods, and all participants provided written verification of consent to participate.

## Imaging technique

BOLD-MRI was performed using a 3.0T magnetic resonance machine (Magnetom Verio, Siemens Medical Solutions, Erlangen, Germany) with 8-channel phased-array coils and 8 gradient-recalled-echoes. Scanning parameters were as follows: acquisition time (TA), 28 seconds; number of slices, 4; slice thickness, 5.0 mm; repetition time (TR), 100 milliseconds; echo delay time (TE), 3.38, 8.23, 12.99, 17.75, 22.51, 27.27, 32.03, and 37.43 milliseconds; flip angle, 60˚; field of view, 350 mm; voxel size, 1.4×1.4×5.0 mm. When the patient's general condition was good, we performed 30% oversampling to improve image quality and reduce artifacts.

**Table 2. Comparison of demographics between study group participants.**

| Parameter | Renal artery stenosis group | Control group |
|---|---|---|
| No. of kidneys | 67 | 60 |
| Average age (years) | 57.63 ± 15.42 | 57.38 ± 14.50 |
| Gender (male/female) | 39/28 | 42/18 |
| SBP(mmHg) | 142.4±11.8 | 140.9±9.7 |
| DBP(mmHg) | 73.3±8.5 | 72.6±7.7 |
| Scr(μmol/L) | 95.6±13.4 | 93.1±13.0 |
| eGFR(mL/min) | 72.03±12.7 | 74.6±12.2 |
| Urine ACR(mg/g) | 115.6±15.5 | 103.7±13.8 |

Urine ACR, Urine Albumin/Urine Creatinine Ratio.

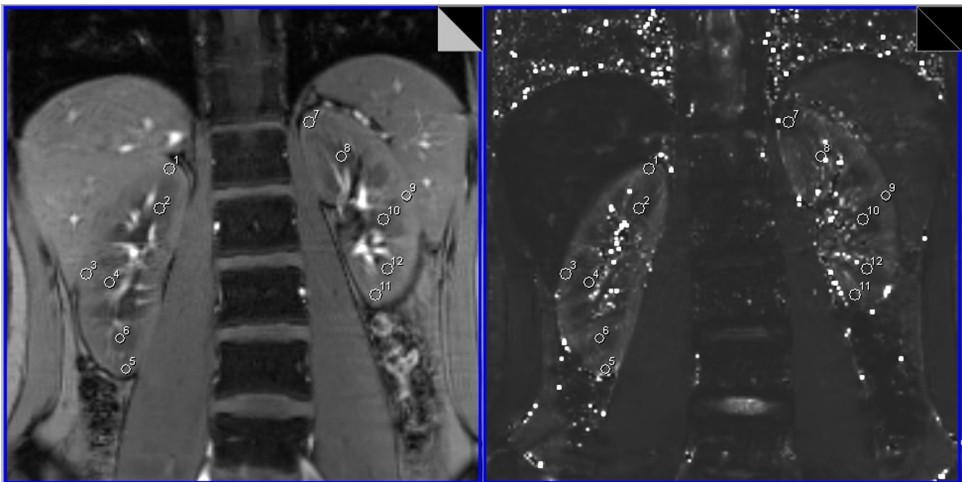

**Fig 1. Regions of interest (ROIs) in the original blood oxygen level–dependent magnetic resonance image, which showed a clear anatomical structure after pairing with the corresponding T2\* mapping image.** ROIs were projected automatically to the T2\* mapping image.

## Post-scan processing

After scanning was completed, we selected an oblique coronal slice with a clear demarcation of the renal cortex and medulla as seen on the Siemens workstation. We used 8 images of the slice with different TE values to create a T2\* mapping image. We then imported the original BOLD-MRI images and the T2\* mapping image into MATLAB(version 7.14; MathWorks, Beijing, China), searching out the BOLD-MRI image of the first TE slice (3.38 ms), which corresponded to the T2\* mapping image. Next, we paired them and drawn the regions of interest (ROI) in both renal cortex and medulla manually on the original BOLD-MRI image, which showed a clear anatomical structure. At the same time, the ROI was projected automatically to the T2\* mapping image to obtain the T2\* value (milliseconds) of the ROI. Then MATLAB provided the reciprocal of T2\* values and converted the units into seconds at the same time. Finally, the R2\* value (1 per second) of the ROI was obtained. Six ROIs were delineated in each kidney, 3 ROIs in the cortex and 3 ROIs in the medulla, located in the upper pole, lower pole and hilum level of the kidney, respectively (Fig 1).

## Statistical analysis

Statistical analysis was performed with using SPSS software 20.0 (SPSS, IBM, Armonk, NY, USA). $P$ values $< 0.05$ were considered to be statistically significant. Paired sample t-test was used to analyse and compare the R2\* values of renal cortex and medulla in the study and control groups. One way analysis of variance (ANOVA) was used to compare the R2\* values in different positions (upper, middle and lower segments) between the study and control groups. The R2\* values in the renal cortex and medulla among the 4 RAS subgroups were analysed and compared with one way ANOVA.

## Results

We performed BOLD-MRI for 40 patients with RAS and for 30 healthy volunteers. Data for 2 patients were excluded because bilateral renal atrophy produced an obscure demarcation between the cortex and the medulla, making R2\* values difficult to measure. Data for another 2 patients were excluded because the patients could not hold their breath during scanning,

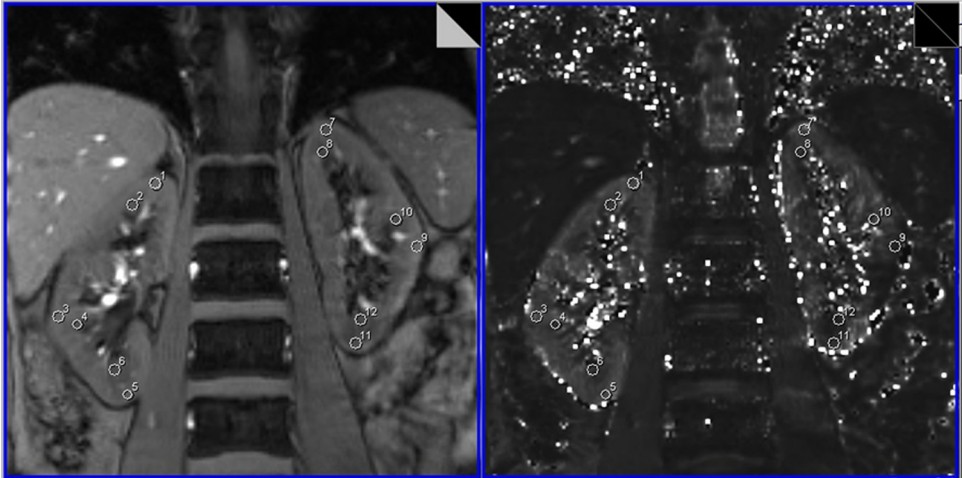

**Fig 2. A 51-year-old woman with a severe renal artery stenosis of left kidney.** ROIs were drawn in the original BOLD image and T2* mapping simultaneously. T2* mapping showed multiple low-signal regions in the left kidney. R2* values (relaxation rates) for the patient were high.

resulting in poor quality images. The remaining 36 patients and 30 volunteers successfully underwent examination with no discomfort (Fig 2).

After all exclusions, we had data for 127 kidneys: 67 in the RAS group and 60 in the control group. BOLD-MRI data are shown in Table 3 (control group) and Table 4 (RAS group).

Data for the 4 subgroups of the RAS group are shown in Table 5.

The mean medulla R2* value was significantly higher than the mean cortex R2* value ($P$ <0.001), regardless of the study or control groups. The mean cortex and medulla R2* values in the RAS group was significantly higher than the mean value in the control group ($P$ = 0.002, and 0.000, respectively).

Comparison of R2* values in the control group showed no statistically significant differences between segments of the renal cortex (upper, middle, and lower segments; $P > 0.05$) or between segments of the renal medulla ($P > 0.05$). The mean R2* value for the upper segment

**Table 3. R2* values for the control group.**

| Kidney layer | Kidney segment | | | Average |
|---|---|---|---|---|
| | **Upper** | **Middle** | **Lower** | |
| Cortex | 18.35 ± 2.35 | 18.49 ± 2.75 | 17.86 ± 3.24 | 18.23 ± 1.77 |
| Medulla | 29.55 ± 3.34 | 29.35 ± 3.46 | 29.93 ± 3.08 | 29.61 ± 2.26 |

Abbreviation: R2*, relaxation value.

**Table 4. R2* values for the renal artery stenosis group.**

| Kidney layer | Kidney segment | | | Average |
|---|---|---|---|---|
| | **Upper** | **Middle** | **Lower** | |
| Cortex | 23.14 ± 8.44 | 20.20 ± 5.07 | 20.08 ± 6.59 | 21.14 ± 4.90 |
| Medulla | 36.99 ± 9.66 | 34.60 ± 8.54 | 37.15 ± 9.38 | 36.25 ± 8.04 |

Abbreviation: R2*, relaxation value.

**Table 5. R2\* value analysis for subgroups with different degrees of renal artery stenosis.**

| Degree of stenosis | Kidney layer | Kidney segment | | | Average |
|---|---|---|---|---|---|
| | | **Upper** | **Middle** | **Lower** | |
| No obvious stenosis | Cortex | 20.10 ± 6.21 | 18.92 ± 4.66 | 17.86 ± 3.23 | 18.96 ± 3.62 |
| | Medulla | 30.97 ± 7.02 | 28.39 ± 4.06 | 30.26 ± 5.63 | 29.87 ± 3.92 |
| Mild stenosis | Cortex | 20.90 ± 3.44 | 19.32 ± 2.66 | 20.38 ± 1.43 | 20.20 ± 2.01 |
| | Medulla | 31.84 ± 5.65 | 31.18 ± 2.36 | 36.43 ± 5.82 | 33.15 ± 2.42 |
| Moderate stenosis | Cortex | 21.48 ± 3.92 | 17.88 ± 2.09 | 18.05 ± 2.49 | 19.14 ± 1.86 |
| | Medulla | 32.05 ± 3.80 | 29.96 ± 3.86 | 33.92 ± 7.53 | 31.98 ± 4.28 |
| Severe stenosis | Cortex | 26.9 ± 11.01 | 22.53 ± 6.70 | 22.67 ± 9.48 | 24.06 ± 5.94 |
| | Medulla | 45.58 ± 8.01 | 42.69 ± 7.47 | 44.34 ± 8.61 | 44.20 ± 6.01 |

Abbreviation: R2\*, relaxation value.

of the renal cortex in the RAS group was significantly higher than the mean values for the middle and lower segments ($P = 0.032$ and 0.025, respectively). No significance difference was found between mean R2\* values for the middle and lower segments of the renal cortex ($P = 0.928$). There was no significant difference in mean R2\* values for different segments (upper, middle, and lower) of the renal medulla in the RAS group ($P > 0.05$).

The mean R2\* value of the upper, middle and lower segments of the renal cortex in patients without obvious stenosis to moderate stenosis was significantly lower than the mean value for patients with severe stenosis ($P = 0.014–0.031$).

The mean R2\* values of the renal cortex in patients without obvious stenosis and with moderate stenosis were significantly lower than the mean value for patients with severe stenosis ($P = 0.001$ and 0.008, respectively). No statistical difference was found in other comparisons of values for the renal cortex.

Regarding the upper segment of the renal medulla with different degrees of RAS, the mean R2\* values for patients without obvious stenosis, those with mild stenosis, and those with moderate stenosis were significantly lower than the mean value for patients with severe stenosis ($P = 0.000$ for all). The same results were found in the renal medulla in the middle segment ($P = 0.000$ for all) and lower segment ($P = 0.000$, 0.024, and 0.001, respectively). For the whole medulla, the result of comparison of the average R2\* value is consistent with the that of comparison of mean R2\* values for the separate segments of the medulla ($P = 0.000$ for all). No statistically significant differences were found in other comparisons for segments of the renal medulla in the RAS group.

Regarding both the cortex and the medulla, the mean R2\* values for patients with RAS of different degrees (i.e., for all subgroups) were significantly higher than the mean R2\* value for the control group ($P < 0.05$), except for the subgroup without obvious stenosis. This result is consistent with the overall results regarding the cortex and medulla values for the RAS group.

## Discussion

In this study we performed a series of comparisons with the following important findings: There was no significant difference between R2\* values for patients with mild or moderate RAS and for patients without obvious stenosis. However, there was a significant difference between R2\* values for the mild and moderate RAS subgroups and the R2\* values of the control group. For the renal medulla, R2\* values for the severe stenosis subgroup were significantly higher than those for the subgroups without obvious stenosis to moderate stenosis. The R2\* values for mild and moderate stenosis subgroups were significantly higher than R2\* values

for the control group. For the renal cortex, the R2* values for the RAS group were significantly higher than the R2* values for the control group. Yet the same differences in medulla R2* values between the severe stenosis and mild/moderate stenosis subgroups were not found in the renal cortex. These phenomena show that BOLD-MRI can detect differences between R2* values for patients with different degrees of stenosis and R2* values for the control group in both the cortex and the medulla. That indicates that BOLD-MRI can reveal the state of oxygenation for both the cortex and medulla in the event of RAS. In addition, BOLD-MRI can distinguish differences in R2* values between states of stenosis except in those cases without obvious stenosis, and it demonstrates that hypoxia of the renal medulla in severe stenosis is worse than in mild or moderate stenosis. This capacity also demonstrates that BOLD-MRI is more sensitive in the medulla than in the cortex.

The basic principle of BOLD-MRI stems from paramagnetic effect of deoxygenated hemoglobin, which can make the local magnetic field inhomogeneous and thus make it possible for the deoxygenated hemoglobin to function as an endogenous contrast agent in imaging. In 1990, Ogawa proposed use of the BOLD contrast method on the basis of this principle. This method makes use of changes in proportions of oxyhemoglobin and deoxyhemoglobin at different values for oxygen partial pressure, which causes changes in $T_2$-weighted image signal intensity, which reflects the oxygenation state of local tissue [9]. The increase in deoxyhemoglobin causes dephasing of protons, shortening of the $T_2$ transverse magnetization vector, a decrease in $T_2$-weighted image signal intensity, and an increase in the R2* value ($1/T_2$). Therefore, a higher R2* value indicates a worsening regional oxygenation state, whereas a low R2* value indicates good oxygenation. With the elimination of internal environment influences such as temperature, pH value, and hematocrit, the BOLD-MRI signal intensity depends mainly on two aspects: the source of oxygen (i.e., the perfusion and diffusion of renal blood flow) and oxygen consumption (i.e., the oxygen consumption rate of the tubular Na-K-ATP enzyme) [10]. The renal blood flow in patients with RAS will decrease proportionally to the degree of stenosis, and as the source of oxygen decreases, the BOLD-MRI signal intensity and R2* value will increase.

In a healthy kidney, oxygen pressure of the cortex is about 50 mmHg and the oxygen pressure of the medulla is between 10 and 20 mmHg. Accordingly, the medulla R2* value should be higher than the cortex R2* value. This phenomenon has been confirmed by the research done by Eckerbom *et al.* [11] on BOLD-MRI in healthy human kidneys. Our findings are consistent with theirs. In addition, we found that both cortex and medulla R2* values for patients with RAS were significantly higher than R2* values for a control group. In another study, Juilliard *et al.* [12] artificially created porcine RAS and found that the R2* values for the renal cortex and medulla in that model were higher than normal. That similar finding demonstrated that RAS decreased renal blood flow that eventually leads to renal hypoxia.

In our study, we divided the RAS group into different subgroups according to degrees of stenosis and compared the R2* values between subgroups for both the cortex and the medulla. We then compared each subgroup with the control group. Our findings demonstrated that the mean medulla R2* value for patients with severe RAS was significantly higher than the mean values for patients with no to moderate stenosis. Regarding the renal cortex, the mean R2* values for the subgroup with severe stenosis were higher than those for the subgroups without obvious and with moderate stenosis. In contrast, the mean R2* values in the medulla for the subgroup with severe stenosis were higher than those for the subgroups without obvious to moderate stenosis. This phenomenon might be related to the low number of participants in the subgroup with mild stenosis group. In the 3 remaining subgroups (no obvious mild and moderate stenosis), the mean R2* values for the renal cortex and medulla were not significantly different.

Both Textor *et al.* [13] and Gloviczki *et al.* [14] reported increased R2* values in patients with RAS when they used BOLD-MRI. Textor *et al.* also found low R2* values in patients with renal artery occlusion and renal atrophy. In addition, Gloviczki *et al.* found that cortical and medullary oxygenation is affected to a lesser degree in patients with moderate or mild RAS. However, until now, there had been no comparisons between healthy persons and patients with different degrees of RAS. Our findings are consistent with those of Textor and Gloviczki *et al*, but our study offers more information by comparing results from the control group with those from the subgroups of patients with varying degrees of stenosis. Thus, findings of this study add valuable information to the current literature.

In addition, in comparing variations in cortex R2* values with variations in medulla R2* values, we found that the increase in medulla R2* values ($\triangle$R2* value = 6.64/s) is greater than that in cortex R2* values ($\triangle$R2* value = 2.91/s). Among the RAS subgroups, the medulla R2* value in severe stenosis was significantly higher than the corresponding value in mild or moderate stenosis, but there was no similar result for cortex R2* values in the different subgroups. These two findings reflect the same phenomenon in ischemia and hypoxia: the changes in BOLD-MRI signal intensity are greater in the renal medulla than in the renal cortex. This phenomenon may be related to characteristics of oxygen dissociation. In the oxygen dissociation curve, when the oxygen partial pressure in the renal cortex is >40 mmHg, most of the hemoglobin is in an oxygenated state. Changes in oxygen partial pressure in this range would not cause a large amount of the oxyhemoglobin dissociation. Therefore, BOLD-MRI is not sensitive to detect changes in oxygen partial pressure in this range. Normal oxygen partial pressure in the renal medulla is between 10 and 20 mmHg, so in that case, a minor change in oxygenation will cause a large alteration in the proportion of deoxyhemoglobin, ultimately leading to a significant change of BOLD-MRI signal intensity [15]. Besides, there is a great deal of Na-K-ATP enzyme in the renal medulla to maintain the balance of sodium. The enzyme's oxygen consumption is close to 50% of full renal oxygen consumption [16], and hypoxia and the heavy load conditions of the renal medulla make it susceptible to hypoxic injury. Because of this, BOLD-MRI can detect R2* value changes in the cortex and medulla, allowing for a more sensitive evaluation of renal medullary oxygenation [17].

The main disadvantage of BOLD-MRI is that many factors can affect local tissue oxygenation state, and these factors may cause changes in the BOLD-MRI signal. These factors include oxygen supply, oxygen consumption, blood flow volume [18], hematocrit [19] and pH value. Therefore, excluding these factors before the examination and considering the influence of different factors during analysis are keys to optimal interpretation of the BOLD-MRI signal. In drawing the ROI manually, identification and avoidance of artifacts is also very important to avoid affecting the accuracy of R2* values. The magnetic susceptibility artifact in BOLD-MRI is usually displayed as a low-signal intensity in an echo image and a high-signal intensity in an R2* mapping [20]. Motion artifacts mainly occur in the upper pole of kidney, and most of them are bilaterally symmetric and thus easily distinguished. When artifacts exist in a ROI, the R2* value or its standard deviation increase unconventionally, possibly reaching a level of 10 or more times the true R2* value. When this occurs, the ROI should be redrawn. R2* values can differ when equipment with different field intensities are used. The higher the field intensity, the higher R2* value [21], and this kind of change is more obvious in the renal medulla. In addition to confirming that BOLD-MRI is more sensitive for detecting changes in renal medullary oxygenation, this phenomenon also suggests that using an R2* value variation as an observed index may produce results that are more significant and accurate than those that are obtained by using an absolute value of R2* in judging oxygenation state and changes. Therefore, seeking a formula to calibrate R2* values for different field strengths may make it possible to obtain uniform R2* values despite equipment dissimilarities.

## Study limitations

Our study has some limitations. First, because of relationships between the RAS and hypertension, most patients in the RAS group had hypertension. We did not exclude the influence of hypertension on R2* values. Second, although we obtained interesting results regarding the changes in R2* values for patients with RAS, there is still much to be done to increase our sample size. Future studies should include a larger cohort to elucidate the differences between subgroups of patients with varying degrees of stenosis.

## Conclusion

BOLD-MRI can effectively evaluate renal oxygenation in patients with RAS. In patients with severe RAS, BOLD-MRI signal intensity changes are more obvious than those with no to moderate stenosis. BOLD-MRI is more sensitive for detecting changes of oxygenation state in the renal medulla than in the renal cortex, thus it can be used as a reliable tool to assess renal ischemia and hypoxia in patients with RAS.

## Author Contributions

**Conceptualization:** Long Zhao, Jiayi Liu.

**Data curation:** Long Zhao, Guoqi Li, Fanyu Meng.

**Formal analysis:** Long Zhao.

**Investigation:** Long Zhao.

**Methodology:** Long Zhao.

**Project administration:** Long Zhao, Jiayi Liu.

**Supervision:** Jiayi Liu.

**Writing – original draft:** Long Zhao.

**Writing – review & editing:** Fanyu Meng, Zhonghua Sun, Jiayi Liu.

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
