## [Decision Letter · Decision Letter 0]

6 Oct 2021

PONE-D-21-22838Cortical and medullary oxygenation evaluation of kidneys with renal artery stenosis by BOLD-MRIPLOS ONE

Dear Dr. Liu,

Thank you for submitting your manuscript to PLOS ONE. After careful consideration, we feel that it has merit but does not fully meet PLOS ONE’s publication criteria as it currently stands. Therefore, we invite you to submit a revised version of the manuscript that addresses the points raised during the review process.

We look forward to receiving your revised manuscript.

Kind regards,

Xi Chen

Academic Editor

PLOS ONE

Journal Requirements:

2. Please ensure that you include a title page within your main document. You should list all authors and all affiliations as per our author instructions and clearly indicate the corresponding author

Reviewers' comments:

Reviewer's Responses to Questions

**Comments to the Author**

1. Is the manuscript technically sound, and do the data support the conclusions?

Reviewer #1: Partly

Reviewer #2: Yes

2. Has the statistical analysis been performed appropriately and rigorously? 

Reviewer #1: Yes

Reviewer #2: Yes

3. Have the authors made all data underlying the findings in their manuscript fully available?

Reviewer #1: Yes

Reviewer #2: Yes

4. Is the manuscript presented in an intelligible fashion and written in standard English?

Reviewer #1: Yes

Reviewer #2: Yes

5. Review Comments to the Author

Reviewer #1: The authors are presenting renal BOLD MRI data in patients with renal artery stenosis and a healthy control group. They report higher R2* values in patients with RAS and also a variation with severity. They have excluded very severe stenosis (close to obstruction) based on earlier reports showing actually lower R2* values. The authors would benefit referencing to recent consensus based technical recommendations [PMID: 31768797].

Specific comments:

1. The major concern with the analysis is the use of small ROIs. While it may work in cortex, it is challenging in medulla due to compromised contrast in patients. In fact, earlier investigators [13, 14] moved to whole parenchyma analysis for this reason [PMID: 22183077]. Figure 1 only shows a small cortical ROI. Even though the legend claims there is good structural information, the image does not show adequate cortico-medullary differentiation. This raises a serious concern about the values reported for the medulla.

2. The use of T2* maps for analysis and reporting R2* values in confusing. In Figure 1, mean value shown is for R2* while Max and Min values are T2*. The authors should simply report T2* values.

3. Not sure if ultrasound based stenosis estimates are accurate. While ultrasound may be used it is usually to perform Doppler ultrasound measurements.

4. The exclusion of diabetics is interesting and not clear if it is valid.

5. Suggesting T2* as phase coherence mapping is misleading. T2* is well established and there is no need to provide further descriptors.

6. Even though 4 slices were acquired, only one slice data was used for analysis. Could you justify?

7. Use of two custom Matlab codes to analyze T2* seems to make no sense. There are several freely distributed analysis programs such as Image J or FireVoxel that could be utilized. The authors should use histogram analysis rather than depend on medullary ROIs [PMID: 22183077]. Alternately and preferably, they should use the TLCO method [PMID: 27798200].

8. While comparison of regional differences in control group may be interesting to demonstrate no significant difference, similar comparison in patients is not justified [Tables 4, 5]. It is quite possible that they also have distal vascular disease leading to more regional differences. These are not necessarily related to the renal artery stenosis.

9. While BOLD-MRI is able to distinguish levels of oxygenation, it is not yet clear it can identify Hypoxia per se. The authors should eliminate the statement “BOLD-MRI can reveal the state of hypoxia” [page 10, line 2].

Minor comments:

10. Either say Medulla R2* and Cortex R2* OR Medullary R2* and Cortical R2*. The authors have repeatedly used Medullar R2* and Cortex R2*.

11. I am not sure if Renal artery stenosis is considered as peripheral artery disease.

Reviewer #2: In this manuscript, Long Zhao et al demonstrated that the mean R2* values for patients with RAS of different degrees were significantly higher than the mean R2* value for the control group both in the cortex and medulla. This result provided us with BOLD-MRI as a reliable tool to assess renal ischemia and hypoxia in RAS. I think authors well prepared this paper and did steady work. This manuscript adequately covered introductions and methods. Interpretation for their results and conclusions seems appropriate.

I have no further comments and endorse this manuscript for publication.

6. PLOS authors have the option to publish the peer review history of their article (what does this mean?). If published, this will include your full peer review and any attached files.

Reviewer #1: No

Reviewer #2: **Yes: **Min Li

---

## [Author Response · Author response to Decision Letter 0]

12 Nov 2021

I’m very grateful for the reviewers' affirmation and further suggestions of our research. The reviewer has mentioned a recent consensus based technical recommendations published in Nov. 2019, named as Consensus‑based technical recommendations for clinical translation of renal BOLD MRI [PMID: 31768797]. We conducted our research from 2017 to 2018, and finished our paper in 2019. We did not refer to the recommendations in our experiment. But I read that review carefully after the reviewer recommend it to me. Fortunately, we keep pace with these top researchers. After a comprehensive survey, they presented final recommendations on renal BOLD MRI data acquisition, analysis and interpretation including 21 items. I check up on our research and find that we reach consensus on at least 17 out of 21 items (81%). The rest of items including other “red light” topics with a 50/50% split in researchers will be useful for us to draw perspectives and design further experiments. Here are the answers to each Specific point raised by the reviewers below：

1. There are several methods of ROI selection in BOLD MRI: the small manual ROI, the large manual ROI, the hybrid ROI and the whole parenchyma analysis method (the compartmental method). Firstly, contrast-enhanced CT images, as used in the hybrid method, are usually unavailable in human research, while their coregistration to MRI is time consuming and requires skill. Secondly, the large ROI method is more vulnerable than the small ROI to partial volume effect of blurry cortico-medullary border. Finally, earlier investigations revealed that the mean R2* values obtained by different ROI methods are close [PMID: 22183077]. The whole parenchyma analysis method has an advantage in comparative studies of pre- to post-drug, because of less time consuming than the manual method, especially in coregistration. But in oxygenation evaluation of kidneys with renal artery stenosis (RAS), we didn’t need to observe the response of R2* value to any drug or intervention. On the contrary, the whole parenchyma analysis method may be sensitive to abdominal susceptibility artifact (ASA), while the manual ROI method could remove it obviously. There were some other reasons made us abandoned the whole parenchyma analysis method, such as its statistical nature. When the amount of available data was limited, the histogram curve was not smooth enough, which led the accuracy might be compromised.

2. The ROIs were drawn on the original BOLD-MRI image, which showed a clear anatomical structure. Then, the ROIs were projected automatically to the T2* mapping image to obtain the T2* value (milliseconds) of the ROI. Then we got R2* values (the reciprocal of T2* values) by program 2. So all the values we shown in results of the paper are R2* values, including mean values, max and min values in Figure 1, just as the consensus‑based technical recommendations recommended. T2* values were just presented in procedure.

3. Each patient underwent renal artery ultrasound examination in the research. Some of them underwent computed tomography angiography (CTA) or magnetic resonance angiography (MRA) as well. But others could only afford the ultrasound examination because of avoiding of radiographic contrast nephropathy. We have tried our best to make the decision of RAS degree accuracy via combined different methods of renal artery examination. Besides, we made a discussion about different methods of renal artery examination in our earlier paper, Comparative study of the diagnostic value of non-enhanced magnetic resonance angiography and ultrasound in the degree of renal artery stenosis in patients with renal insufficiency (Doi:CNKI:SUN:XFXZ.0.2021-03-012.) .

4. The mechanism of R2* values variation of diabetics is not clear. We are planning to study it specifically in next research. This is the reason we excluded diabetics in this research.

5. I followed reviewer’s advice and deleted the redundant description.

6. We did acquired 4 slices, but only used only one slice data for each patient. Just as reviewer mentioned in question 1, adequate cortico-medullary differentiation is necessary for ROI selection. But some oblique coronal slices which far from the renal hilus could not obtain adequate cortico-medullary differentiation leading to volume averaging. So we selected a slice with a most clear demarcation of the renal cortex and medulla out of 4 slices to drawn ROIs. Consequently, we got less data as well as more accurate data and more sensitive results. Benefitting of reviewer’s advice, we have noticed that more slices are recommended [PMID: 22183077]. We would consider this point in the further research.

7. We use two custom Matlab codes in post-scan processing for different aims. Get the TE slice (3.38 ms) by code 1. Draw ROIs and obtain R2* values by code 2. Corresponding simplifications have been completed in manuscript. We used Image J in the preliminary study. The usage of TLCO method and histogram analysis has been discussed in answer 1.

8. Comparison of regional differences shown some positive results in patients. It does be quite possible that they also have other disease leading to more regional differences. We are also curious that how does variation of regional R2* values occurred in RAS patient. We are considering to do further study in these patients or distal renal artery stenosis patients.

9. Corresponding modifications have been completed in manuscript.

10. Corresponding modifications have been completed in manuscript.

11. Renal artery stenosis may be not considered as peripheral artery disease. Corresponding modifications have been completed in manuscript.

1. Some of data of this study are not publicly available due to privacy or ethical restrictions. Because raw data contain patients’ names, IDs, genders, ages, medical histories, diseases etc.

Some of data of this study cannot be shared because the data also forms part of an ongoing study. 

2. Data requests could connect to the academic research office of Bejing Anzhen hospital. 

E-mail: anzhenkjc@163.com

---

## [Decision Letter · Decision Letter 1]

4 Jan 2022

PONE-D-21-22838R1Cortical and medullary oxygenation evaluation of kidneys with renal artery stenosis by BOLD-MRIPLOS ONE

Dear Dr. Liu,

Thank you for submitting your manuscript to PLOS ONE. After careful consideration, we feel that it has merit but does not fully meet PLOS ONE’s publication criteria as it currently stands. Therefore, we invite you to submit a revised version of the manuscript that addresses the points raised during the review process.

We look forward to receiving your revised manuscript.

Kind regards,

Xi Chen

Academic Editor

PLOS ONE

Reviewers' comments:

Reviewer's Responses to Questions

**Comments to the Author**

1. If the authors have adequately addressed your comments raised in a previous round of review and you feel that this manuscript is now acceptable for publication, you may indicate that here to bypass the “Comments to the Author” section, enter your conflict of interest statement in the “Confidential to Editor” section, and submit your "Accept" recommendation.

Reviewer #1: (No Response)

2. Is the manuscript technically sound, and do the data support the conclusions?

Reviewer #1: Partly

3. Has the statistical analysis been performed appropriately and rigorously? 

Reviewer #1: Yes

4. Have the authors made all data underlying the findings in their manuscript fully available?

Reviewer #1: Yes

5. Is the manuscript presented in an intelligible fashion and written in standard English?

Reviewer #1: Yes

6. Review Comments to the Author

Reviewer #1: The authors did respond adequately to many of the critical issues identified in the prior version. However, the most critical one remains unresolved (#1 below).

Specific comments:

1. It is acceptable to use small ROIs, but then we need a clear illustration of the cortical and medullary ROIs. Figure 1 shows only one cortical ROI. The legend suggests that this belongs to the same data shown in Figure 2. The authors should combine them as one figure. Show all the ROIs drawn on both the anatomical image and the T2* map. They should do the same with Figure 3 by including the anatomical image and showing all the ROIs. The T2* maps should include a gray scale bar and they should use the same values for both data sets so that we can appreciate differences in T2* values.

2. Prior critique, “In Figure 1, mean value shown is for R2* while Max and Min values are T2*.” The authors responded saying the Max and Min values are R2*. How is it mathematically possible to have Min value 58 and Max value 60 and have a mean value 16.9492. Clearly the 16.9492 1/s equals 59.17 ms which is consistent with the Max and Min values shown.

3. The choice of reporting only one slice out of four acquired slices is not convincing. If the choice is based cortico-medullary contrast, the data shown in Figure 1 is disappointing.

4. Figure 4 shows only medulla R2* values. Need for this figure is not clear given that the data is included in Table 5.

7. PLOS authors have the option to publish the peer review history of their article (what does this mean?). If published, this will include your full peer review and any attached files.

Reviewer #1: No

---

## [Author Response · Author response to Decision Letter 1]

12 Jan 2022

There are still several issues which need to be settled after the prior version. Here are the answers to each specific point raised by the reviewers below：

1. A combined image showing all the ROIs of the anatomical image and the T2* map is very helpful, which could improve the legibility and reliability of the figure. So we have modified “Figure 1 and 2” into “Figure 1” with the anatomical image, the T2* map and all the ROIs. Absolutely, we have done the same with “Figure 3” by including the anatomical image and showing all the ROIs, name as “Figure 2”, just as the reviewer instructed. A T2* map with a gray scale bar is also an inspirational idea, but the R2* values of many organizations in the whole FOV will be extremely out of range, leading to an extremely large scale, and the intuitive readability of the scale bar will be poor. There is an illustration in attached files "Response to Reviewers".

2. In the “Figure 1” of prior version, “mean value shown is for R2* while Max and Min values are T2*” do exist, we have update new figures with all the ROIs including anatomical image and T2* map.

3. Changes in renal parenchymal oxygenation caused by renal artery stenosis are sometimes only local. At this time, if the T2* values are measured and averaged by the whole kidney, the changes of T2* values in local renal tissue may be covered up. On the contrary, it is possible to miss lesions when measuring T2* value with multiple small ROIs, so we try to value the upper, middle and lower parts of the kidney separately to avoid missing lesions. The two measurements require a balanced strategy. The choice of reporting only one slice out of four acquired slices might miss lesion which located in other slices, but differences between groups have already discovered by middle slices. Meanwhile, the cortical-medulla boundary of the kidney is clearer in the middle layer, and at the same time, the partial volume effect of the marginal layer is avoided, resulting in inaccurate T2* value.

4. We remove “Figure 4” because of medulla R2* values shown in “Figure 4” were already included in Table 5, just as the reviewer mentioned.

---

## [Decision Letter · Decision Letter 2]

15 Feb 2022

Cortical and medullary oxygenation evaluation of kidneys with renal artery stenosis by BOLD-MRI

PONE-D-21-22838R2

Dear Dr. Liu,

We’re pleased to inform you that your manuscript has been judged scientifically suitable for publication and will be formally accepted for publication once it meets all outstanding technical requirements.

Kind regards,

Xi Chen

Academic Editor

PLOS ONE

Additional Editor Comments (optional):

Reviewers' comments:

Reviewer's Responses to Questions

**Comments to the Author**

1. If the authors have adequately addressed your comments raised in a previous round of review and you feel that this manuscript is now acceptable for publication, you may indicate that here to bypass the “Comments to the Author” section, enter your conflict of interest statement in the “Confidential to Editor” section, and submit your "Accept" recommendation.

Reviewer #1: All comments have been addressed

2. Is the manuscript technically sound, and do the data support the conclusions?

Reviewer #1: Yes

3. Has the statistical analysis been performed appropriately and rigorously? 

Reviewer #1: Yes

4. Have the authors made all data underlying the findings in their manuscript fully available?

Reviewer #1: Yes

5. Is the manuscript presented in an intelligible fashion and written in standard English?

Reviewer #1: Yes

6. Review Comments to the Author

Reviewer #1: Authors have addressed the prior critical issues. Even though the explanation for not including a gray scale bar is not convincing, it is not as important.

7. PLOS authors have the option to publish the peer review history of their article (what does this mean?). If published, this will include your full peer review and any attached files.

Reviewer #1: No

---

## [Editor Report · Acceptance letter]

28 Feb 2022

PONE-D-21-22838R2 

Cortical and medullary oxygenation evaluation of kidneys with renal artery stenosis by BOLD-MRI 

Dear Dr. Liu:

I'm pleased to inform you that your manuscript has been deemed suitable for publication in PLOS ONE. Congratulations! Your manuscript is now with our production department. 

Kind regards, 

on behalf of

Dr. Xi Chen 

Academic Editor

PLOS ONE